# Primary and Metastatic Intracranial Ewing Sarcoma at Diagnosis: Retrospective International Study and Systematic Review

**DOI:** 10.3390/cancers12061675

**Published:** 2020-06-24

**Authors:** Lianne M. Haveman, Andreas Ranft, Henk van den Berg, Stephanie Klco-Brosius, Ruth Ladenstein, Michael Paulussen, Heribert Juergens, Uta Dirksen, Johannes H.M. Merks

**Affiliations:** 1Princess Maxima Center for Pediatric Oncology, 3584 CS Utrecht, The Netherlands; J.H.M.Merks@prinsesmaximacentrum.nl; 2Department of Pediatric Hematology and Oncology, University Children’s Hospital of Essen, Center of Pediatric Oncology, 45122 Essen, Germany; andreas.ranft@uk-essen.de (A.R.); Stephanie.Klco-Brosius@uk-essen.de (S.K.-B.); Uta.Dirksen@uk-essen.de (U.D.); 3Department of Pediatric Oncology, Academic Medical Center, Emma Children’s Hospital, 1105 AZ Amsterdam, The Netherlands; h.vandenberg@amc.uva.nl; 4Children’s Cancer Research Institute, 1090 Vienna, Austria; ruth.ladenstein@ccri.at; 5Vestische Kinder- und Jugendklinik, Witten/Herdecke University, 45711 Datteln, Germany; M.Paulussen@kinderklinik-datteln.de; 6Department of Pediatric Hematology and Oncology, University Children’s Hospital, 48149 Muenster, Germany; jurgh@ukmuenster.de

**Keywords:** Ewing sarcoma, intracranial, metastatic, outcome

## Abstract

Intracranial Ewing sarcoma (EwS) is rare and publications on primary or metastatic intracranial EwS are minimal. The aim of this study was to describe incidence, clinical behavior, treatment, and factors associated with outcome in patients with primary intracranial EwS or patients with a primary extracranial EwS and cerebral metastases at diagnosis. We reviewed all patients with primary or with metastatic intracranial EwS at diagnosis registered in the International Clinical Trial Euro-E.W.I.N.G.99 (EE99). In total, 17 of 1435 patients (1.2%) presented with primary intracranial EwS; 3 of them had metastatic disease. Four patients (0.3%) with primary extracranial EwS presented with intracranial metastatic lesions. The 3-year event-free survival (EFS) was 64% and overall survival (OS) was 70% in patients with a primary intracranial EwS. Local control in patients with primary intracranial EwS consisted of surgery (6%), radiotherapy (RT) (18%), or both modalities (76%). Univariate analysis showed that patients < 15 years of age had significantly better outcome (EFS: 72%; OS: 76%) compared to those aged above 15 years (EFS: 13%; OS: 25%). In conclusion, primary intracranial EwS and extracranial EwS with cerebral metastases at diagnosis is rare, yet survival is comparable with local and metastatic EwS elsewhere in the body. Age and stage of disease are important prognostic factors. Besides chemotherapeutic treatment, local control with surgical resection combined with RT is recommended whenever feasible.

## 1. Introduction

According to the World Health Organization, the peripheral primitive neuroectodermal tumors, classic Ewing Sarcoma (EwS) and extraskeletal EwS, constitute the Ewing sarcoma family of tumors. Primitive neuroectodermal tumors (PNETs) are composed of small undifferentiated embryonal-type round cells. For intracranial PNETs, a distinction is made between primitive neuroectodermal tumors of the central nervous system which includes medulloblastoma, supratentorial PNETs, and pineoblastoma, versus primitive neuroectodermal tumors of the peripheral nervous system, which encompasses Ewing sarcomas (EwSs), mainly arising from bone and soft tissue. The distinction between primitive neuroectodermal tumors of the central nervous system and peripheral primitive neuroectodermal tumors, or intracranial EwSs, is important regarding clinical behavior, treatment, and prognosis [1,2,3]. Although EwSs of the central nervous system may histologically appear quite similar to primitive neuroectodermal tumors of the central nervous system, EwSs show the characteristic chromosomal translocation t(11;22)(q24;12) in 85–90% of patients. This translocation results in fusion of the gene EWSR1 with the ETS family gene FLI1 (alias EWSR2). Less frequently, in about 10% of patients, EwSR1 fusions with other members of the ETS family occur, most commonly ERG. Occasionally, fusions between EwSR1 and non-ETS gene family members are seen [4,5]. These translocations are not found in primitive neuroectodermal tumors of the central nervous system. In addition, immunochemistry is used to distinguish intracranial EwSs from primitive neuroectodermal tumors of the central nervous system. Immunostaining for CD99, encoded by the MIC2 gene, is usually positive in EwSs of the central nervous system and negative in most primitive neuroectodermal tumors of the central nervous system. However, this antigen is not specific and is also expressed in some other central nervous system (CNS) tumors [5,6].

In children, the incidence of EwS is about 4.5 cases/million a year, with a peak incidence of 11 cases/million at the age of 12 years [7]. Primary intracranial EwS is extremely rare and the number of publications on primary intracranial EwS or patients with extracranial EwS with cerebral metastases is minimal; nearly all are confined to single case reports. Adverse prognostic factors in EwS are age > 14 years, initial tumor volume > 200 mL, poor histological response to induction chemotherapy, male gender, high lactate dehydrogenase (LDH) at diagnosis, and metastatic disease [8,9]. Whether these factors are relevant in intracranial EwS is unknown. To sort this out, we reviewed all patients with primary intracranial EwS and patients with extracranial EwS and intracranial metastatic disease at diagnosis treated according to the International Clinical Trial Euro-E.W.I.N.G. 99 (EE99) protocol and included in the GPOH-EE99 registry [10]. The aim of this study was to describe the incidence, clinical behavior, and treatment received, and analyze the factors associated with outcome in patients with primary intracranial EwS and patients with primary extracranial EwS and cerebral metastases at diagnosis. In addition, we performed a literature review on previously published reports on this topic. Therewith, this paper encompasses the largest series on intracranial EwS published in the literature to date. 

## 2. Results

### 2.1. Patient Characteristics and Treatment

In total, 1435 patients were included in the EE99 database during a 10-year period. The median follow-up was 4.2 years (range 0.87–12.68). In total, 21 patients (=1.5%) presented with a primary or metastatic intracranial EwS. Seventeen patients (=1.2%) presented with primary intracranial EwS. Three of these 17 patients, in addition to their intracranial primary, had metastatic lesions in the CNS. Four out of 21 patients (=0.3%) were diagnosed with an extracranial primary EwS elsewhere in the body and had intracerebral metastatic lesions. In Table 1, the characteristics and treatment modalities of these patients are shown. In seven patients, all with primary intracranial EwS, cerebrospinal fluid (CSF) puncture was performed; malignant cells were found in two patients. 

The median age of patients with primary intracranial EwS at diagnosis was 11.1 years (range 3.6–69.3 years); 12/17 (71%) of patients were younger than 18 years. The median age of the 4 patients who presented with intracranial metastatic disease was 16.4 years (range 14.1–53.1 years). In 16/17 patients with primary intracranial EwS, the tumor volume at diagnosis was known. In 13/16 patients, the tumor volume was below 200 mL. In 3 of the 4 intracranial metastatic patients, the primary tumor volume was above 200 mL. 

In eight patients, the clinical symptoms at diagnosis could be extracted from physician’s letters. Clinical symptoms were rather diverse and consisted of headache and double vision (patient number (pat.nr. 1); epileptic seizures and behavior change (pat.nr. 2); headache and hyposmia (pat.nr. 5); headache and swelling (pat.nr. 8); swelling without further symptoms (pat.nr. 10, pat.nr. 16); headache and papilledema (pat.nr. 12); and fever, general malaise, and swelling (pat.nr. 21; localization iliac bone with occipital brain metastasis). Sixteen of 21 patients had cranial bone involvement. 

All patients received induction chemotherapy with vincristine, ifosfamide, doxorubicin, and etoposide (VIDE) according to the EE99 protocol. Two patients with primary intracranial EwS underwent high-dose chemotherapy (HDCT) followed by autologous stem cell transplantation (aSCT) in the consolidation phase; one of them had proven EwS cells in the CSF at diagnosis. Before HDCT, this patient received three courses of vinblastine combined with intrathecal etoposide (pat.nr. 7). The other patient (pat.nr. 17), who also had a (smaller) lesion in the sacrum, received HDCT after eight courses with vincristine, actinomycine D, and ifosfamide (VAI). According to EE99, patients with metastatic disease were to receive HDCT followed by aSCT.

However, one patient with extracranial EwS and metastatic intracerebral lesions showed progressive disease after the fourth VIDE course and antitumor treatment was stopped. One patient with primary intracranial EwS (pat.nr. 15) received intrathecal topotecan, although this patient had a negative CSF puncture. 

In four of the five patients with a CNS primary without bone involvement, upfront surgery was performed. In two additional patients, upfront surgery was done, both because of symptoms due to increased intracranial pressure. In all other patients, local therapy was performed after induction chemotherapy. In 13/17 (76%) of the patients with primary intracranial EwS, local therapy consisted of both surgery and radiotherapy (RT). In 3/17 patients (18%), definitive RT was given; two of these patients had intracerebral metastases. In only 1/17 patients (6%) who presented with primary EwS of the brain, surgery without additional RT was performed. In this patient, surgery was radical and there was a good histopathological response. Three of the four patients with extracranial EwS with metastatic intracranial disease received definitive RT; the fourth patient showed progression of disease before local therapy could be started.

For 13/14 patients who underwent surgery, data on resection margins were known. In 3/13 patients, a radical resection was achieved; in 4/13, a marginal resection; and in 6/13, the resection was intralesional. Histological response was reported in only four patients; two had a poor histological response.

During follow-up, 3 of 17 patients with primary intracranial EwS (17.6%) had a local recurrence (LR) after initial complete remission; intracranial relapse occurred in one of the four patients with extracranial EwS and cerebral metastases. Two of the 17 patients (11.7%) with primary intracranial EwS showed progressive disease while on therapy, as did 2/4 patients with intracerebral metastatic disease at diagnosis. 

### 2.2. Outcome and Prognostic Factors

After a median follow-up time of 4.2 years, the 3-year event-free survival (EFS) for the total cohort was 0.57 (standard error (SE) = 0.11) and 3-year overall survival (OS) was 0.62 (SE = 0.11). In the cohort of patients with primary intracranial EwS, 11 patients were free of disease (65%); 3-year EFS was 0.64 (SE = 0.12) (Figure 1a) and OS was 0.70 (SE = 0.11) (Figure 1b). The three patients with primary intracranial EwS and CNS metastases at diagnosis did worse; only one was alive after 9.7 years of follow-up. For patients with extracranial EwS with intracerebral metastatic disease at diagnosis, only one patient was alive after 9.3 years. All patients with LR (*n* = 4) and the patients with progressive disease under therapy (*n* = 4) died. Univariate analysis was performed on patients with primary intracranial EwS in order to determine the influence of local treatment modalities on survival. 

However, numbers in the subgroups were not equally distributed. Due to small numbers, no statistical analysis was performed; however, a descriptive analysis of the relation between therapy and survival showed no difference in survival between treatment with RT alone or combined surgery and RT. Three patients (18%) received RT alone; two of them had brain metastases. Two patients were alive after 3.8 and 4.0 years of the follow-up period. One patient with CNS metastases died after 1.1 years of treatment. Thirteen patients (76%) received combined local treatment. Eight patients were alive after a median follow-up time of 8.1 years. The single patient treated by surgery only was alive after 7.86 years of follow-up. This patient had a surgical resection with negative margins and a good histological response to chemotherapy.

In 12/17 patients, the resection margins were known. In the patients with surgical resection and negative margins (*n* = 2), no events occurred. In both the group with marginal resection (*n* = 4) and the group of patients with an intralesional resection (*n* = 6), all received RT; half of the patients were alive at the end of the follow-up period. Studying the patients with primary intracranial EwS, no difference in EFS could be found between the 12 patients with involvement of cranial bones (3-year EFS 0.66 (SE = 0.14)) compared to the 5 patients without bone involvement (EFS 0.60 (SE = 0.14)). The 3-year OS was 0.74 (SE = 0.13) and 0.60 (SE = 0.22), respectively (*p* = 0.80). Patients with an age below 15 years (*n* = 11) had a 3-year EFS of 0.81 (SE = 0.12) compared to an EFS of 0.17 (SE = 0.15) for patients above 15 years (*n* = 6) years of age (*p* = 0.015). The 3-year OS was 0.76 (SE = 0.16) compared to an OS of 0.33 (SE = 0.19), respectively (*p* = 0.05).

Only one of the three patients with a tumor volume >200 mL was alive after 4 years of follow-up (EFS 0.33; OS 0.33). In the 13 patients with a tumor volume <200 mL, EFS was 0.68 and OS 0.66, respectively. Due to the limited number of patients with a large tumor volume, statistical analysis was not possible. 

A multivariate analysis of prognostic factors was not performed due to the limited sample size.

### 2.3. Literature Review

The literature search revealed a total of 920 articles. The process of the publication retrieval and the inclusion and exclusion of the studies was performed according to the Preferred Reporting Items for Systematic Reviews and Meta-analyses (PRISMA) guidelines and is shown in Figure 2 [11]. By reviewing the titles and abstracts, seven studies on primary intracranial Ewing sarcoma describing three or more cases were considered [12,13,14,15,16,17,18]. No studies with three or more patients with primary extracranial EwS and intracerebral metastatic disease at diagnosis could be included. Details of the seven included studies describing patients with primary intracranial EwS are described in Table 2. Due to heterogeneous studies, a meta-analysis of the studies was not possible.

In conclusion, the 7 selected studies were very dissimilar and included only small patient numbers in the range of 3–14 patients. Only two studies [13,15] included patients treated according a uniform chemotherapy protocol, with respectively four and seven patients. Important data on local treatment [14,15,16,18] and/or outcome [17,18] were missing in most of the studies. In general, the authors suggest that long-term survival may best be achieved if an Ewing-based chemotherapy regimen is given, and (radical) surgery is combined with RT [12,13,14].

## 3. Discussion

Intracranial EwS is rare, and meticulous delineation from other intracranial primitive neuroectodermal tumors has to be assured. Of all patients included in the EE99-trial, 17 patients (1.2%) presented with a primary intracranial EwS with involvement of the dura or cerebral parenchyma. Three of them in addition had CNS metastases at diagnosis. This is in line with reported incidences of primary intracranial EwS of 1–6% in earlier studies [19,20]. In our study, only four patients (0.3%) with a primary extracranial EwS presented with intracerebral metastatic lesions at diagnosis. This is the largest cohort treated according to a uniform chemotherapeutic regimen to date. In the literature, no other cohort studies could be found describing patients with extracranial EwS and intracerebral metastases at diagnosis. However, intracranial metastases at a later stage of the disease, as a result of hematogenous metastatic spread, is more frequently reported in the range of 2–10% [21,22,23].

In patients with intracranial EwS, the disease often originates from the cranial bones. Usually, this also means involvement of the meninges and the epidural space [13,24,25]. EwSs that do not originate from the cranial bones are described to a lesser extent. Recently, however, a patient with a sellar/suprasellar mass with intraventricular extension has been described [26].

In a previous report, we described 51 patients with EwS of the head and neck region included in the EE99-trial [27]. In this study, 23 of the 51 of the patients were diagnosed with a primary EwS of the skull. In the present study, only patients with lesions within the cerebrum or entering the cerebrum or infiltration of the dura were included. 

Treatment options in patients with cerebral EwS are identical to EwS elsewhere in the body and include chemotherapy, surgery, and/or RT. Patients who underwent surgery and/or RT combined with chemotherapy had better outcomes as compared to patients who did not have adjuvant chemotherapy [12,13,14,25]. The efficacy of chemotherapy could be limited by the ability of the drugs to cross the blood–brain barrier (BBB). In the EE99 protocol, patients are all treated with induction chemotherapy with vincristine, ifosfamide, doxorubicin, and etoposide (VIDE). Ifosfamide is assumed to penetrate through the blood–brain barrier due to its lipid solubility, small molecular size, and minimal binding to plasma and tissue proteins. This is confirmed in several studies; however, interpatient variability of CFS/plasma ratios is described, which may explain differences in efficacy and toxicity among patients [28,29]. Limited data are available about the penetration of etoposide across the BBB. For etoposide, the tissue to blood ratios seem low [30,31]. The penetration for doxorubicin and vincristine into the brain parenchyma is also low [32,33]. In the presented study cohort, one patient received intrathecal topotecan in addition to the VIDE chemotherapy backbone, although the CSF at diagnosis was negative for malignant cells in this case. Another patient received three courses of vinblastine combined with intrathecal etoposide, scheduled shortly before high-dose chemotherapy. Both patients were alive at follow-up. The value of intrathecal chemotherapy in EwS of the CNS is unknown; however, the National Cancer Institute as part of the Intergroup Ewing’s Sarcoma Study Committee treated a group of 93 EwS patients with prophylactic CNS irradiation and intrathecal methotrexate. It was concluded that the risk of CNS involvement in patients with EwS was not affected [23]. 

The survival rate in EwS has improved due to the implementation of multi-modality treatment. OS is about 70% in patients with localized disease; however, OS is lower than 30% in patients with metastatic disease [8,34]. In this study, the 3-year EFS was 64% and OS was 70% in patients with primary intracranial EwS. This is in line with the survival rate of localized EwS elsewhere in the body. Consistent with earlier studies, in our study, patient age below 15 years and localized disease seemed to be good prognostic factors [8,9]. Involvement of bone structures did not seem to affect outcome. Patients with metastatic disease had a 3-year EFS and OS of 28%, comparable with results reported earlier [25]. 

In general, treatment of EwS includes at least marginal surgical resection. Surgery has to be combined with RT in the case of narrow margins and/or poor histological response. Surgical resection with negative margins in the case of intracranial tumors is very difficult to achieve in most of the patients; therefore, RT should be added. In order to limit unacceptable morbidity from surgery, a less invasive resection in tumors that involve the skull base and vascular regions, such as the cavernous sinus, may be accepted [35]. In patients with EwS of the head and neck region, no difference could be found in the outcome between patients treated with surgery, RT, or combined surgery and RT [27]. In the present study, 76% of the patients received combined modality local treatment. There seems to be no difference in outcome between local treatment groups, but definitive conclusions cannot be drawn due to the small patient numbers. Limitations of this study include also its retrospective nature. However, based on the available data, a treatment composed of chemotherapy, surgery combined with RT is advised according to the general guidelines for treatment of EwS. However, in patients with intracranial EwS, radical surgery will not be feasible in a considerable proportion of patients. In these patients, and in patients with metastatic disease, definitive RT should be given. Stereotactic radiosurgery seems a promising treatment modality, but no experience is reported in intracranial EwS.

## 4. Materials and Methods 

### 4.1. Study Population

We analyzed data from Ewing sarcoma patients, included in the German Society for Pediatric Oncology and Hematology (GPOH) database of the EE99 trial between September 1999 and September 2009 [10]. Patients with primary intracranial EwS or patients with primary extracranial EwS and cerebral metastases at diagnosis were included in the study. If available, detailed information was extracted from medical letters and reports or patient files. Patients were included if the lesions were located within the cerebrum or entering the cerebrum, i.e., with infiltration of the dura. Patients with lesions outside the cranial bones were only included if there was also an intracerebral component with involvement of the cerebral parenchyma. Patients were excluded for this study if there was an intracranial lesion without involvement of the dura and/or cerebral parenchyma. Primary EwS was confirmed by histopathology and molecular diagnostics in all patients. Patients less than 50 years of age were eligible and included in the EE99 study. Patients older than 50 years and patients who did not meet inclusion criteria or fulfilled exclusion criteria were included as registry patients. Informed consent was obtained from all patients and/or legal guardians. The protocol (ClinicalTrials.gov identifier: NCT00020566) was reviewed and approved by the appropriate institutional review boards, ethical committees, and legal authorities. Informed consent was obtained from all patients and/or legal guardians according to the Declaration of Helsinki and national guidelines. 

### 4.2. Treatment

EE99 induction chemotherapy consisted of 6 courses of VIDE. Local control was generally carried out after the sixth course of induction chemotherapy, with a preference for surgical intervention with or without additional RT. Following local therapy, risk-adapted chemotherapy was administered as part of the randomized study questions [10]. After surgery, the histopathological response was determined. A good histological response was defined as less than 10% of viable tumor cells in the specimen.

### 4.3. Survival

Event-free survival (EFS) was defined as the interval between the date of diagnosis and the date of the first event. In the absence of events, patients were censored on the date of their most recent consultation. An event was defined as progressive disease, relapsed disease (local or metastatic), secondary malignancy, or death from any cause. Overall survival (OS) was defined as the time from diagnosis until death from any cause. 

### 4.4. Statistical Analyses

Statistical analyses were performed with SPSS statistics 22 (IBM Corporation, Armonk, NY, USA) and SAS 9.2 (SAS Institute, Cary, NC, USA) software packages. Event-free survival (EFS) and overall survival (OS) were calculated using the Kaplan–Meier method. Surviving patients were censored at the date of last contact. Univariate comparisons were estimated using the log-rank test. 

### 4.5. Literature Review

A PubMed database search for primary intracranial Ewing sarcoma and extracranial EwS with intracranial metastases was performed. We used a Medline search strategy with words from the title or the abstract:

#1 for Ewing sarcoma: Ewing OR ewings OR ewing* OR ewing sarcoma OR peripheral neuroectodermal tumor OR peripheral neuroectodermal tumour OR peripheral neuroectodermal tumors OR peripheral neuroectodermal tumors OR primitive neuroectodermal tumor OR primitive neuroectodermal tumor OR primitive neuroectodermal tumors OR primitive neuroectodermal tumors OR Primitive Neuroepithelial Tumor OR Primitive Neuroepithelial Tumors OR Primitive Neuroepithelial Tumor OR Primitive Neuroepithelial Tumors OR Primitive Neuroepithelial Neoplasm OR Primitive Neuroepithelial Neoplasms OR PNET OR PNETs OR askin. 

#2 for primary intracranial or intracranial metastases: intracranial OR cerebral OR CNS OR dural A search until 31 December 2019 was conducted. For this study, searches were limited to articles published in English and only studies describing 3 or more patients were included. To identify additional eligible studies, reference lists of included studies were screened.

## 5. Conclusions

Primary and metastatic intracranial EwS at diagnosis is very rare. This study shows that survival in both primary intracranial and metastatic disease is comparable to local and metastatic EwS elsewhere in the body. Known prognostic factors in patients with EwS as age, disease stage at diagnosis, and also tumor volume seem to also hold for patients with intracranial EwS. In general, patients with intracranial EwS should, if the clinical symptoms permit, be treated according to a standard EwS treatment, where induction chemotherapy is followed by local therapy and subsequently maintenance chemotherapy. For local control, surgery combined with RT is recommended. However, no differences in survival could be detected comparing surgical outcome in respect to the achieved margins. Patients that received definitive RT showed comparable survival and is indicated in patients where surgery is not possible or in patients with metastatic disease. 

## Figures and Tables

**Figure 1 cancers-12-01675-f001:**
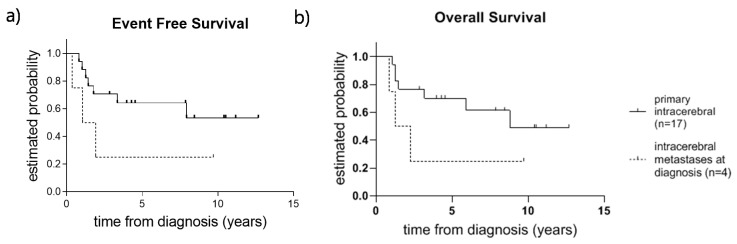
(**a**) Event-free survival (EFS) and (**b**) overall survival (OS) in patients with primary intracranial EwS (*n* = 17) and extracranial EwS patients with intracerebral metastases at diagnosis (*n* = 4).

**Figure 2 cancers-12-01675-f002:**
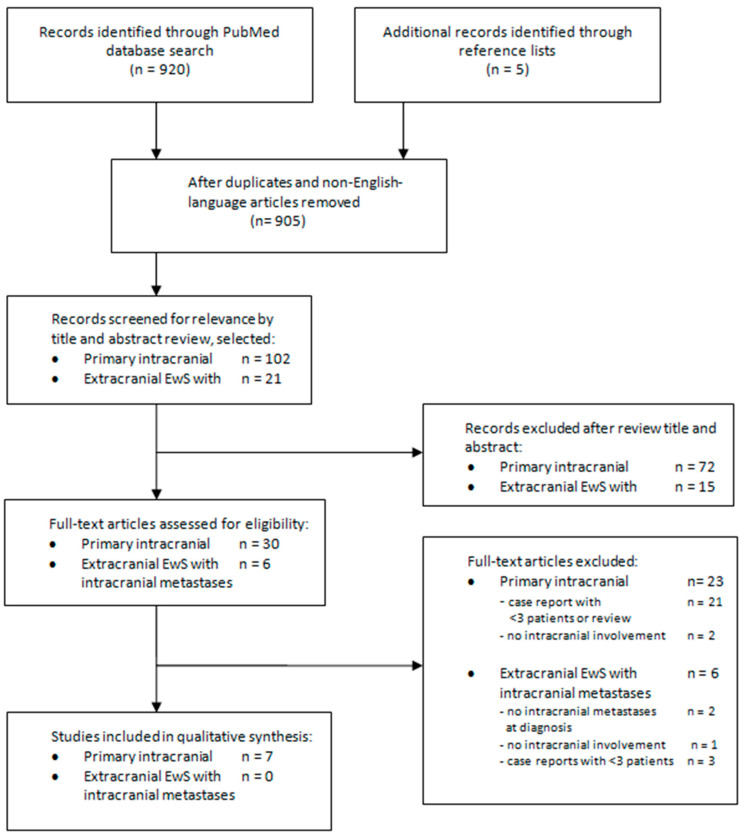
The process of publication retrieval, and the inclusion and exclusion of studies.

**Table 1 cancers-12-01675-t001:** Baseline characteristics and treatment modalities of included patients with primary and metastatic intracranial Ewing sarcoma.

Pat. Nr.	Age (Years)	Sex (M/F)	Tumor Volume >200 mL	Meta-Stases (M): CNS, B, BM, P	Origin	Cranial Bone Involvement (yes/no)	-Histopathology -Immunohistochemy -Translocation	Local Therapy(SR and/or RT)	Radical/Marginal/ Intra-Lesional Surgery	Good (<10% Vital Cells) or Poor HR	Chemo-Therapy Courses	Pr. or LR	Patient Status (Dead/Alive)	OS (Year)	EFS (Year)
**Primary intracerebral**															
1	11.08	F	No (117 mL)	No	Frontal lobe	yes	Neurodiff. PAS+, CD99+ S100+, NSE+ Desmin+ synapth+	SR and RT	marginal	NR	6x VIDE 8x VAI		alive	8.45	8.45
2	53.76	M	No	Yes: CNS, L+	Temporal	no	Neurodiff. S100+	Upfront resection + RT	Intra-lesional	UR	6x VIDE 7x VAC	Pr	died	1.26	1.02
3	12.78	M	No	No	Posterior cranial fossa	yes	Neurodiff. CD99+, S100− Mol: t(11,22) neg	SR and RT	Intra-lesional	NR	6x VIDE 1x VAI 7x VAC		died	5.90	3.35
4	19.72	F	No	No	Skull base left/intra-cranial	yes	Undiff.	RT	-	-	6x VIDE 1x VAI 7x VAC		alive	8.79	7.93
5	30.36	F	Yes	No	Frontodural	yes	Neurodiff. Pas+, mic+, s100+ vimentin+	SR and RT	Intra-lesional	Poor HR	6x VIDE 8x VAI	LR	died	3.16	1.78
6	7.44	M	No (176 mL)	No	Frontobasal	no	Neurodiff.	SR and RT	Intra-lesional	Poor HR	6x VIDE 6x VAI 2x VAC		alive	11.18	11.18
7	9.63	M	Yes (236 mL)	Yes: CNS, L+	Parietal, M in cere-bellum	yes	Neurodiff. Mol: t(11,22) neg; 22q12	RT	-	-	6x VIDE3 x other * 1x HD		alive	3.97	3.97
8	6.95	M	No	No L-	Fronto-parietal; epidural and dural	yes	Neurodiff. Pas+, mic+, nse− negatief, s100? Vimentin+	-Upfront resection -pre-op. RT +SR of resttumor	-intra-lesional -radical	-UR -Good HR	6x VIDE 1x VAI 7x VAC		alive	10.4	10.4
9	15.71	F	No	No	Para-meningeal, area middle cranial fossa	yes	Undiff.	SR and RT	marginal	NR	6x VIDE 1x VAI 7x VAC	LR	died	1.48	1.43
10	6.08	F	No	No	Frontal lobe	yes	Neurodiff. MIC2+, S100+ NSE+ vimentin+	SR	radical	Good HR	6x VIDE 8x VAI		alive	7.86	7.86
11	10.48	M	No (67 mL)	No	Temporal	yes	Neurodiff. Mol: t(11,22) pos	SR and RT	radical	NR	6x VIDE 1x VAI 7x VAC		alive	4.29	4.29
12	69.34	M	No	No	Parieto- occipital	no	Neurodiff. CD99+, NSE+ vimentin+ Mol: t(11,22) pos	Upfront resection + RT	marginal	UR	6x VIDE 7x VAC	LR	died	1.26	1.26
13	10.30	F	No (23 mL)	No	Fossa posterior	no	Undiff. PAS+, CD99+ S100− Vimentin+ desmin− Mol: no t(11,22) but breakage of EwS gene on 22	Upfront resection + RT	Intra-lesional	UR	6x VIDE 8x VAI		alive	10.52	10.52
14	3.58	F	No (86 mL)	No L-	Temporal	yes	Neurodiff. PAS+, CD99+, NSE+ vimentin+ desmin− Mol: (t11,22) neg Ews/fli 7/5 (type 2)	SR and RT	Intra-lesional	NR	6x VIDE 8x VAC		alive	2.84	2.84
15	4.36	F	No (22 mL)	No L-	Parietal	yes	Neurodiff. PAS+, CD99+, S100+, desmin− Mol: chr22q12	Upfront resection + RT	marginal	UR	6x VIDE + other ** 8x VAC		alive	4.53	4.53
16	33.46	F	?	No	Occipital with intra-cerebral lesions	no	Neurodiff. PAS+ CD99+, S100+ NSE+ desmin−	Upfront resection + RT	radical	UR	6x VIDE 8x VAI		alive	12.68	12.68
17	11.85	M	Yes (291 mL)	Yes: B, CNS, L?	Parietal + os sacrum ***	yes	CD99+ S100− desmin− Mol: Ews/Fli 7/6 type I; t(11,22)	RT	-	-	6x VIDE 8x VAI 1x HD	Pr	died	1.08	0.81
**Patients with extracranial EwS and intracerebral metastases at diagnosis**															
18	16.55	M	Yes (628 mL)	Yes: CNS, B, BM	Tibia; multifocal skull, intra-cerebral, and meningeal	yes	Undiff. PAS− CD99+	RT		-	6x VIDE 1x VAI 1x HD	Pr	died	1.26	1.05
19	14.12	F	Yes	Yes: CNS, B, BM	costal; frontal epidural	yes	Mol: t(11,22) transcript 22Q12	RT		-	6x VIDE 1x VAI 1x HD		alive	9.70	9.70
20	53.13	F	Yes	Yes: CNS, P	sacrum; fossa posterior	yes	Undiff. No t(11,22) or t(21,22)	no		-	4x VIDE	Pr	died	0.87	0.36
21	16.21	M	No (137 mL)	Yes: CNS	os ilium; occipital	yes	Neurodiff. Mol: t(11,22) pos	RT		-	6x VIDE 8x VAI 1x HD	LR	died	2.52	1.92

CNS = central nervous system, RT = radiotherapy, SR = surgical resection, EFS = event-free survival, OS = overall survival, CNS = central nervous system, P = pulmonary, B = bone, BM = bone marrow, NR = not reported, UR = Upfront resection, HR = histological response, VIDE = vincristine, ifosfamide, doxorubicin, and etoposide, VAI = vincristine, actinomycin D, ifosfamide; VAC = vincristine, actinomycin D, cyclophosphamide; HD = high dose chemotherapy; Pr = progression under treatment; LR = local recurrence. * = three courses of vinblastine combined with intrathecal etoposide; ** = intrathecal topotecan; *** large intracerebral lesion, small lesion sacrum, unknown, which lesion was the primary lesion.

**Table 2 cancers-12-01675-t002:** Characteristics and treatment modalities of studies with case series of three or more patients with primary intracranial Ewing sarcoma.

Author	No. of Subjects	No. of Patients with Metastatic Disease	Mean Age (Range) in Years	Mean Follow-up (range) in Months	Uniform Chemo-Therapy Protocol	No. of Patients Treated with CT	No. of Patients with SR	No. of Patients with Radical SR	No. of Patients with Marginal SR	No. of Patients with RT alone	No. of Patients with SR and RT	LR Rate (%)	Survival
**Primary intracerebral**													
Chen et al. [12]	14	3	14.1 (1–43)	30.1 (6–84)	No	10	14	7 (50%)	7 (50%)	-	9	71.4	5-year OS 19%
Colak et al. [13]	4	-	13.8 (6–26)	32.0 (11–69)	Yes	4	4	3 (75%)		1	3	NR	100%: mean 32 months (11–69)
Yang et al. [14]	4	-	10 (5–16)	49.5 (12–126)	No	4	4	NR	NR	-	4	25%	25% died (36 months) 75% alive: mean 54 (12–126) months
Singh et al. [15]	7	-	13 (7–21)	26.9 (12–48)	Yes	2 (+1 pat. after LR)	7	NR	NR	-	4	14%	57% died, mean 33.2 months PFS *n* = 3: 23.3 months
Jing et al. [16]	8	3	15 (7–23)	NR	NR	NR	8	NR	NR	-	8	NR	0%
Ke et al. [17]	3	-	19.3 (15–28)	NR	Yes	3	3	2	1	-	3	66.7%	*n* = 1: >6 years *n* = 2 with LR: lost to f.u.
Vanden Heuvel et al. [18]	3	-	21.8 (2.4–61)	NR	No	2	3	NR	NR	-	1	-	*n* = 1: >5 years *n* = 1: >18 months *n* = 1: lost to f.u

No. = number, SR = surgical resection, RT = radiotherapy; LR = local recurrence; EFS = event-free survival, OS = overall survival; PFS = progression-free survival; NR = not reported, P = progression.

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
