# Peer review of "Primary and Metastatic Intracranial Ewing Sarcoma at Diagnosis: Retrospective International Study and Systematic Review"

_cancers, 2020, doi:10.3390/cancers12061675_

Round 1

Reviewer 1 Report

This is a review of patients treated on the Euro-Ewing 99 trial that had primary intracranial Ewing sarcoma or extracranial Ewings with intracranial metastases at diagnosis. The authors identify 17 patients with primary intracranial Ewing sarcoma, resulting in the largest series describing this patient population in the published literature. They also conduct a systematic review of prior evidence describing this patient population. The manuscript shows that these patients have similar overall and event free survival to extracranial Ewing sarcoma patients and demonstrates the importance of multimodality therapy with chemotherapy, radiation therapy, and surgery in these patients. The study is reasonably designed, and the conclusions are supported by the data. I recommend the following revisions of the manuscript prior to publication:

Major comments:

It needs to be clarified in the abstract, introduction, and throughout the manuscript that the four patients with intracranial metastases at diagnosis were patients who had extracranial primary sites of disease or intracranial disease. It is confusing whether we are talking about intracranial Ewings with CNS metastases or extracranial Ewings with CNS metastases in places. When I was initially reading the manuscript, I thought the authors were talking about intracranial Ewings metastasizing to other intracranial sites. Please carefully find all locations where these four patients are mentioned in the manuscript and assure that the language is clear.

In the conclusion of the manuscript, the authors state that “radical surgery combined with RT is recommended.” However, the data show that margin negative radical surgery was only possible in a single patient. The remaining patients had marginal or intralesional resections, as is true for most intracranial tumors. Radical surgery therefore needs to be removed from the conclusion.

A radical resection can still result in a positive margin. For example, a radical nephroureterectomy can have a positive margin. In the manuscript between lines 154 and 167 the terms radical resection and margin negative resection seem to be used interchangeably. This needs to be addressed.

Minor comments:

Line 24/Abstract: Please change “Aim of this study…” to “The aim of this study…”

Introduction – The multiple abbreviations cPNET, sPNET, and EwS are hard to follow in this paragraph.  I would suggest simply using the full spelled out versions of central nervous system PNET and supratentorial PNET since they are not used many times in the manuscript.

Line 46-48: I suggest moving the comment about the WHO definition of pNET and EwS to the first sentence of the introduction to immediately orient the reader to the differences in these tumor types. Then transition to how it is molecularly identified.

Line 68/Introduction: Please change “Aim of this study…” to “The aim of this study…”

Line 91/Results: The abbreviation UPN is not defined in the manuscript. After reading the manuscript, this abbreviation corresponds to patient numbers, but it is not defined. Furthremore, in Table 1 patient number is abbreviated as Pat. Nr. I recommend keeping the abbreviation for each and defining it when it is first used in the manuscript.

Line 96/Results: VIDE chemotherapy is not defined in the manuscript at the time of its first use. It is defined in the methods, but since they are at the end of the manuscript it is difficult to know the agents that constitute VIDE when it is first used.

Table 1: I suggest adding “cranial” to bone involvement

Table 1: VIDE, VAI, VAC, HD chemotherapy regimens need to be defined in the table legend

Line 117: Please change “definite RT” to “definitive RT.” Make sure that RT is defined when it is first used in the manuscript.

Line 132: Please make sure that “SE” is defined when first used in the manuscript. Standard error?

Table 2: et all is italicized for Yang et al, but not others. Please change to a consistent format.

Table 2: PFS is used in the entry for Singh et al, but not defined in the table legend

Line 191/Discussion: Please change “In literature…” to “In the literature…”

Line 194-196/Discussion: This sentence needs to be rewritten using improved sentence structure for clarity. It seems the authors are trying to say that intracranial disease is usually (but not always) caused by extension from bones of the cranium.

Line 198/Discussion: Please change “In present study only…” to “In the present study only…”

Line 219/Discussion: Please change “Survival rate in EwS…” to “The survival rate in EwS…”

Line 220/Discussion: Please change “OS is about 70% in patients with localized patients…” to “OS is about 70% in patients with localized disease…”

Line 223/Discussion: Please change “Conform earlier studies…” to “Consistent with earlier studies…”

Line 227/Discussion: Please insert a comma after “In general”

Line 232/Discussion: Please insert a comma after “region”

Line 239-240/Discussion: Please change to “In these patients, and in patients with metastatic disease, definitive RT…”

Author Response

We thank the reviewer for all the valuable comments. Major comments * The reviewer indicates that it is confusing whether we are talking in the article about patients with primary intracranial EwS or about the patient group with extracranial EwS with cerebral metastases at diagnosis. We thank the reviewer for this comment and have clarified this throughout the document (lines 26. 30. 36, 70,71, 89-91, subheading in table 1, lines 123, 134-135, 143, 152, 216, 311. In the conclusion we stated that radical surgery combined with radiotherapy is recommended. The reviewer is right that radical surgery was only reached in a single patient and this is also daily clinical practice. We removed 'radical' from the conclusion; and we now state that surgery combined with radiotherapy is recommended (line 333). In the manuscript (lines 156-160) we now use the terms concerning radicality of surgery and margins in a more consistent way and also changed 'Radical surgery' to ’surgical resection with negative margins' in the Discussion (line 235). Minor comments * We changed line 24/Abstract: from “Aim of this study…” to “The aim of this study…” * Introduction: We removed the abbreviations of cPNET and sPNET, and used the full spelled out words to increase readability and also started the introduction with the WHO definition of Ewing sarcoma family of tumours. * We changed line 68 in the Introduction (line 76 in new version) “Aim of this study…” into “The aim of this study…” * We agree the abbreviation for patient numbers is not used consequently in the Results section and Table 1, we changed this consequently to patient numbers (pat.nr.), both in text and Table. * We explained the abbreviation of VIDE in the Result section, and removed this in the Method section. * Table 1: As suggested, we added “cranial” to bone involvement and defined the VIDE, VAI, VAC, HD chemotherapy regimens in the Table legend. * We changed “definite RT” to “definitive RT.” (line 117, new: line 130) and also defined radiotherapy (RT) (line 130), where it is used for the first time in the manuscript. * We explained the abbreviation SE in line 147, where it was used for the first time. * Table 2: et all in italic for Yang et al, is removed to have a consistent format. * In Table we added the definition of PFS in the table legend. * We changed line 191 (new 209) from “In literature…” to “In the literature…”. * We rewrote the sentence in Line 194-196 (new 211-213) into: "In patients with intracranial EwS, the disease often originates from the cranial bones. Usually this also means involvement of the meninges and the epidural space". * We changed line 198 (new: line 217) “In present study only…” to “In the present study only…” * We changed line 219 (new: line 238) “Survival rate in EwS…” to “The survival rate in EwS…” * We changed line 220 (new: line 239) “OS is about 70% in patients with localized patients…” into “OS is about 70% in patients with localized disease…” * We changed line 223 (new: line /Discussion: Please change “Conform earlier studies…” into “Consistent with earlier studies…” * We inserted a comma after "In general" (Line 227, new: line 246). * We inserted a comma after "region" (Line 232; new: line 251). * We changed line 239-240 (new “In these patients, and in patients with metastatic disease, definitive RT…”

Reviewer 2 Report

It is a well-written and scientific article about the rare entity primary and metastatic intracranial Ewing's sarcoma. Although the methodology for the systematic review of the literature is analysed, it would be best if you could reform it according to PRISMA guidelines and include the PRISMA outline.

Author Response

Thank you for your suggestion to use the PRISMA guidelines to improve our Methodology. We performed the Pubmed search again, and we described the process of publication retrieval, inclusion and exclusion of the studies in a Figure conform the PRISMA guidelines and outline (Figure 2, line 202). Because included studies were very small and heterogeneous it was not possible to perform a quantitative analysis. Hopefully, this figure will contribute to a clearer methodology of the literature study.

Reviewer 3 Report

The authors report in their paper a review of patients with primary or metastatic intracranial EwS registered in the International Clinical Trial Euro-E.W.I.N.G.99, carrying out a further systematic review of the literature, illustrating and commenting on the updated treatment modalities. The manuscript is detailed and well written, summarizes the state of the art on EwS brain metastases and primary locations, and is interesting and well structured. it is certainly of interest for publication. However, i advice to include in the discussion a comment on the solitary sellar-suprasellar localizations of EwS and how the total surgical removal associated with adjuvant treatments can prolong survival:

- First Case of Primary Sellar / Suprasellar-Intraventricular Ewing Sarcoma: Case Report and Review of the Literature
Mattogno PP et al. World Neurosurg. 2017 Feb; 98: 869.e1-869.e5. doi: 10.1016 / j.wneu.2016.12.045.

Author Response

We thank the reviewer for his comments. As the reviewer suggested, we refer to the article about a very rare presentation of a primary sellar/suprasellar-intraventricular EwS (Mattogno PP et al, First Case of Primary Sellar / Suprasellar-Intraventricular Ewing Sarcoma: Case Report and Review of the Literature. World Neurosurg. 2017 Feb; 98: 869.e1-869.e5. doi: 10.1016 / j.wneu.2016.12.045). In addition to Line 213-216: " In patients with intracranial EwS, the disease often originates from the cranial bones. Usually this also means the involvement of the meninges and the epidural space [13, 23, 24]". We added the sentence: "EwS that do not originate from the cranial bones are described to a lesser extent. Recently, however, a patient with a sellar / suprasellar mass with intraventricular extension has been described [25]". For all the locations of an intracranial EwS local treatment advice remains the same: This means surgical resection, followed by radiotherapy in case of positive margins or definitive RT in those patients where surgery is not possible. This is already stated in line 260-262. To increase the readability of the article, we think we should not repeat this.